# Pharmacokinetics of Curcumin Delivered by Nanoparticles and the Relationship with Antitumor Efficacy: A Systematic Review

**DOI:** 10.3390/ph16070943

**Published:** 2023-06-29

**Authors:** Fernanda Silvestre, Carolina Santos, Vitória Silva, Alicia Ombredane, Willie Pinheiro, Laise Andrade, Mônica Garcia, Thyago Pacheco, Graziella Joanitti, Glécia Luz, Marcella Carneiro

**Affiliations:** 1Laboratory of Bioactive Compounds and Nanobiotechnology (LCBNano), Campus Darcy Ribeiro, University of Brasilia, Brasilia 70910-900, Brazil; 2Post-Graduate Program in Nanoscience and Nanobiotechnology, Institute of Biological Sciences, Campus Darcy Ribeiro, University of Brasilia, Brasilia 70910-900, Brazil; 3Post-Graduate Program in Biomedical Engineering (PPGEB), Faculty of Gama, University of Brasilia, Special Area of Industry Projection A, Brasilia 72444-240, Brazil; 4Department of Nutrition, Faculty of Health Sciences, Campus Darcy Ribeiro, University of Brasilia, Brasilia 70910-900, Brazil; 5Post-Graduate Program in Sciences and Technologies in Health, Faculty of Ceilândia, Campus Darcy Ribeiro, University of Brasilia, Brasilia 72220-275, Brazil; 6Post-Graduate Program in Animal Biology, Institute of Biological Sciences, Campus Darcy Ribeiro, University of Brasilia, Brasilia 70910-900, Brazil

**Keywords:** curcumin, pharmacokinetics, nanoparticles, cancer

## Abstract

Curcumin is a polyphenolic compound, derived from Curcuma longa, and it has several pharmacological effects such as antioxidant, anti-inflammatory, and antitumor. Although it is a pleiotropic molecule, curcumin’s free form, which is lipophilic, has low bioavailability and is rapidly metabolized, limiting its clinical use. With the advances in techniques for loading curcumin into nanostructures, it is possible to improve its bioavailability and extend its applications. In this review, we gather evidence about the comparison of the pharmacokinetics (biodistribution and bioavailability) between free curcumin (Cur) and nanostructured curcumin (Cur-NPs) and their respective relationships with antitumor efficacy. The search was performed in the following databases: Cochrane, LILACS, Embase, MEDLINE/Pubmed, Clinical Trials, BSV regional portal, ScienceDirect, Scopus, and Web of Science. The selected studies were based on studies that used High-Performance Liquid Chromatography (HPLC) as the pharmacokinetics evaluation method. Of the 345 studies initially pooled, 11 met the inclusion criteria and all included studies classified as high quality. In this search, a variety of nanoparticles used to deliver curcumin (polymeric, copolymeric, nanocrystals, nanovesicles, and nanosuspension) were found. Most Cur-NPs presented negative Zeta potential ranging from −25 mV to 12.7 mV, polydispersion index (PDI) ranging from 0.06 to 0.283, and hydrodynamic diameter ranging from 30.47 to 550.1 nm. Selected studies adopted mainly oral and intravenous administrations. In the pharmacokinetics analysis, samples of plasma, liver, tumor, lung, brain, kidney, and spleen were evaluated. The administration of curcumin, in nanoparticle systems, resulted in a higher level of curcumin in tumors compared to free curcumin, leading to an improved antitumor effect. Thus, the use of nanoparticles can be a promising alternative for curcumin delivery since this improves its bioavailability.

## 1. Introduction

Cancer is a group of diseases responsible for approximately 10 million deaths in the world in 2020, and it has been named one of the most devastating malignancies [1].

The limitations of traditional treatments have stimulated research for new therapeutic alternatives to improve the efficacy of the treatments and decrease the adverse effects. In this context, researchers have focused special attention on the application of antitumor compounds from natural sources. For millennia, natural products have been used as a major source of drugs, and nowadays about half of the pharmaceuticals in use are derived from natural products [2]. Phytotherapy has found a place worldwide, with about 35,000 herbal species showing anticancer properties, according to the National Cancer Institute (NCI) in the USA [2]. Phytomolecules have unique advantages, such as low price, fewer adverse effects, and lower toxicity [3].

Curcumin, a polyphenol extracted from the rhizomes of turmeric (*Curcuma longa*), has been reported as a potential alternative therapy, exhibiting antitumor, anti-proliferative, antimetastatic, anti-angiogenic, anti-inflammatory, and antioxidant properties, among many other biological activities. Curcumin, also known as diferuloylmethane, consists of two phenyl rings substituted with hydroxyl and methoxyl groups and connected via a seven-carbon keto-enol linker, with a molecular weight of 368.38. The chemical structure of curcumin is the basis of its biofunctional effects [4,5,6]. Its efficacy in preventing the formation and spread of tumors or reducing their size has been reported for several types of tumors, including stomach, cervix, skin, lung, colorectal, and breast cancers [6,7,8,9]. The antitumor effect of curcumin involves multiple mechanisms, such as the modulation of proapoptotic and antiapoptotic proteins, growth factors (HER-2 and EGFR), enzymes (matrix metalloproteases), inhibition of STAT3 and NF-kB signaling pathways, reduced production of inflammatory mediators (COX-2, lipoxygenase 2, iNOS, and related cytokines), and suppression of angiogenic cytokines (IL-6, IL-23, and IL-1β) [9,10].

Despite all the health benefits, the use of free curcumin as a therapeutic drug is limited due to its hydrophobicity, resulting in poor water solubility and fast metabolism and systemic clearance, especially in the intestine and liver [7,11,12]. To overcome these limitations by increasing curcumin solubility and bioavailability, nanobiotechnology can be helpful [8,13,14].

In vivo studies have reported that nanostructured curcumin (Cur-NPs) promotes a sustained release of Cur and shows better performance against many types of cancer, such as breast, skin, brain [15,16,17], cervical tumor [11,18], lung [14,19], Hodgkin’s lymphoma [20] and colon-rectal [13]. This seems to be due to the reduced opsonization process which leads to increased plasma Cur concentration for a longer time compared to free curcumin [7].

Therefore, the aim of this present review is to bring together research involving the comparison of the pharmacokinetics (biodistribution, bioavailability and clearance) between free and nanostructured curcumin and their respective relationships with antitumor efficacy. State-of-the-art research on Cur-NPs will contribute to the optimization of analyses and development processes of phytochemicals’ nanodelivery systems, aiming to achieve the status of a safe and effective technology for use in oncology practice. The curcumin-loaded nanoparticles could be implemented for overcoming their poor bioavailability and extend their biomedical applications [4].

## 2. Results

### 2.1. Selection of Studies

The string (“curcuma longa” AND nanoparticle AND anticancer AND (cancer OR tumor)) was performed based on the research question, with all synonyms for each term.

Based on this string, 3 studies were found in EMBASE, 35 in Science Direct, 108 in SCOPUS, 91 in Web of Science, 103 in Pubmed and 5 in Clinical Trials, totaling 345 initial studies. Curcumin is a plant widely used in medicine and the number of studies with its biocompounds has been increasing over the years. Research on *Curcuma longa* has a significant number of publications. This can be observed using the search string TITLE-ABS-KEY (‘curcumin’/exp) in the scientific database EMBASE; 24,026 documentary results were found, with the search carried out on 6 December 2022. Searching for the same string in the SCOPUS database, 42,349 studies were obtained, of which some of the regions/countries that most published were China, India, the United States, Iran, and Italy, where China published 21.2 per cent and India 18.9 per cent of the studies, based on the SCOPUS database.

When filtering the search using the string with the search terms (‘curcumin’/exp AND ‘pharmacokinetics’/exp AND ‘antineoplastic agent’/exp) in the EMBASE database, on 6 December 2022, 5445 publications were found. The search presents publications registered since 1978 (Figure 1). It is noted that from 2004 onwards there was a gradual increase in research with curcumin, and in 2020, there was the highest peak of publications with these terms and their synonyms, despite the COVID-19 pandemic.

Using the same search string, 5445 publications were obtained from all the years included, of which 3475 were found in the MEDLINE and EMBASE databases together, 137 only in the MEDLINE database,1832 only in the EMBASE database and 1 publication of Preprints. According to SCOPUS, using the string (curcumin AND pharmacokinetics AND antineoplastic agent”), there were 18,048 document results, where 22.9 per cent represent the subject area of Pharmacology, Toxicology and Pharmaceuticals and 53.2 per cent of the all documents are Articles.

In these studies, a bibliometric analysis was performed using the VOSviewer software version 1.6.17 [21], with the configuration: co-occurring terms in the title and abstract fields, “complete count”, with at least 50 simultaneous occurrences of terms. A relevance score was calculated. Based on this score, the most relevant terms were selected. The default choice was to select the 60 per cent of the most relevant terms. Thus, 614 keywords were found. The result shows the most frequent terms in the works; the term “curcumin” has an occurrence of 5226 times, the term “antineoplastic activity”, 1125 times, and the term “nanoparticle”, 1025 times. It was also observed that the term “pharmacokinetic parameters” presented a low occurrence of only 125 times. This demonstrates that studies on the pharmacokinetics of the use of curcumin in the treatment of cancer are not as recurrent among the studies selected from the EMBASE database, which reinforces the relevance of this review study.

In Figure 2, we see the links between the terms. Note that the term “antineoplastic activity” is related to the term “curcumin”, “drug release”, and “human”, among others. On the other hand, the term “pharmacokinetics” only has a connection with the term “curcumin”. This term, “curcumin”, in turn, presents several correlations of terms, such as “human”, “drug delivery system”, and “nanocarrier”, among others. It is then observed that the terms “antioneoplastic agent”, “pharmacokinetic parameters” and “curcumin” present correlations in the studies of the EMBASE database, based on the search string (“curcumin”/exp AND “pharmacokinetics”/exp AND “antineoplastic agent”/exp). We can visualize the data in Figure 3 according to four different clusters (groups). Each cluster is represented by the main terms linked to the other terms: “Nonhuman”, which is behind the term “curcumin” (yellow), “human” (red), “curcumin” (green), “antineoplastic agent” (blue). The lines between the terms indicate the relationships between them in the different articles analyzed. It can be noted that many of these studies are on the use of *Curcuma longa* in medicine. Some of the most common terms in this field are drug delivery system, drug release, drug bioavailability, animal experiment, article, review etc.

In relation to the selection of studies, in the first stage (“title and abstract” assessment), 243 studies were found after removal of the duplicate. Further, 11 studies were excluded by automation tools. The remaining 232 articles were analyzed by full-text peer review by group members. This process led to the exclusion of 221 articles according to the exclusion criteria (Appendix A). In the end, 11 articles were kept and included in this systematic review. A flowchart detailing this process is shown in Figure 4.

### 2.2. Characteristics of the Included Studies

All the included studies are research articles that evaluated the pharmacokinetics and antitumoral activity of free curcumin (Cur) and curcumin nanoparticles (Cur-NPs) in in vivo models of cancer. Table 1 summarizes the characteristics of the included studies. The selected studies were conducted only in two countries: China (*n* = 9) and the United States (*n* = 2). Cur-NPs used in the included studies were mainly described by hydrodynamic diameter (HD) (*n* = 11), polydispersity index (PdI) (*n* = 9) and zeta potential (*n* = 10) The encapsulation efficiency of curcumin (EE per cent) was evaluated by HPLC (*n* = 5), ultraperformance liquid chromatography (UPLC) (*n* = 1), and spectrophotometry (*n* = 5).

All the pharmacokinetics analyses were evaluated by HPLC using samples of plasma (*n* = 10), liver (*n* = 4), tumor (*n* = 3), lung (*n* = 2), brain (*n* = 2), kidney (*n* = 1), and spleen (*n* = 1). The animal models used were Sprague Dawley rats (*n* = 6), BALB/c (*n* = 4) and Kunming strain (*n* = 1).

In relation to the pharmacokinetics and biodistribution, it was observed that the dose of Cur and Cur-NPs used ranged from 1.25 to 100 mg/kg or 50 µg/mL in one study. Regarding the frequency of treatment, it was observed that almost all studies applied a single injection of Cur or Cur-NPs (*n* = 10) by intravenous (*n* = 9) or via oral (*n* = 2). In one of these studies, the treatment was carried out daily for 14 days. The time of analysis of all studies ranged from 0.5 h to 24 h. In two studies, the plasma was evaluated for 3 [14] or 30 days [15]. In two studies, both the pharmacokinetics and biodistribution analysis were performed for similar times. Other studies only carried out pharmacokinetics or biodistribution analysis (*n* = 9).

Considering the antitumoral activity evaluation, almost all the studies assessed tumor progression (*n* = 10). The dose of nanostructured Cur ranged from 0.8 to 100 mg/kg. Concerning the frequency of treatment, the authors applied a single injection (*n* = 2), daily (*n* = 3), on alternate days (*n* = 5) or 5 times of week (*n* = 1) for 7 to 30 days. It was observed that most studies carried out the treatment of Cur and Cur-NPs by intravenous (*n* = 8) or via oral (*n* = 3).

### 2.3. Synthesis of Results

The included studies (Table 1) used different nanostructures with curcumin, such as a nanovesicular carrier (Exo) [11], four polymeric structures with PGLA (CPN, CPTN, PLGA-NPs, HPB-PLGA NPs) [12,14] and one of chitosan (CLN-CUR) [14], five copolymer structures with MPEG (CUR-NPs, CUR-P-NPs, FA/Nano-CUR, Nano-CUR, CUR-MPEG-PLA) [13,15,19,22], a solid lipid nanoparticle framework (SLN-CUR) and a vitamin E framework (TPGS-CUR) [20], a nanostructure using nanosuspension (CUR-NSps) [7], and two structures with nanocrystals (CUR-NC, CUR-NCHA) [23].

The smallest particle had hydrodynamic diameter (HD) of 30.47 ± 0.65 nm [13], and the largest HD value was 550.1 nm [14]. The smaller one presents the type of structure of copolymer with MPEG (FA/Nano-CUR), and the larger one presents the structure of the nanoparticle of polymer with chitosan (CLN-CUR).

For the polydispersion index (PDI) analysis, a variation from <0.06 [12] to 0.283 ± 0.006 [20] was noted, where for the nanoparticle with the lowest PDI value, its structure was a polymer (CPNT), and the highest PDI value presented the Vitamin E nanostructure (TPGS-CUR). For zeta potential (ZP) analysis, the lowest value found was −25.0 ± 0.8 [23], and the highest was 12.7 mV [14], and the nanoparticles are based on a structure with nanocrystals and polymer with chitosan, respectively. The encapsulation efficiency percentage (EE per cent) ranged from 7.9 per cent to 98.8 per cent.

For the experimental design in the pharmacokinetics and antitumoral evaluations, mice and rats of both sexes were used. In fact, many mouse models have been described, including BALB/c [13,15,19,20,23], Balb/c nude [22], athymic nude mice (*n* = 1) [11], mice of the Kunming strain [12], Sprague-Dawley mice [7,11,15,22,23], C57BL/6 (13,15), ICR mice [7], SCID mice [20]. In the studies both female [11,13,15,20] and male mice [7,12,14,15,19] were used for pharmacokinetic and antitumor analysis (*n* = 11). In addition, there was one study where both pharmacokinetics and the biodistribution evaluation were performed [7]. For the antitumor study, many cancer cell lines were used: CaSki human cervical squamous cell carcinoma [11], Hca-F mouse hepatocarcinoma [12], MCF-7 human breast cancer [22], A-549 human lung adenocarcinoma [9], L-540 Hodgkin’s lymphoma [20], H22 murine hepatocarcinoma [7], CT26 colorectal cancer [13], 4T1 murine mammary carcinoma [23], HeLa human cervical adenocarcinoma [18], H446 human small cell lung cancer [14], B16F10 murine melanoma [16] and A375 human malignant melanoma [16].

Nanostructures with curcumin were evaluated in several animals with different types of cancer. For cervical tumor treatment (*n* = 1), HPB PLGA NPs (18) showed a 75% increase in cervical cancer tumor growth inhibition. For liver cancer (*n* = 3), ExoCur [11] increased the inhibition of cervical tumor growth by 61%, CUR-NSp [7] increased the tumor inhibition rate by 70.34%, 40.3% and 53.21%, respectively, for these dosages and by 56.2% for CPN and 84.4% for CPTN, compared to free curcumin [12] (31.1%). For the treatment of breast cancer (*n* = 2), curcumin-containing nanostructures of CUR-NP and CUR-P-NP, compared to free curcumin, showed that after 2, 6 and 24 h there was an increase in concentration in tumor tissue after intravenous injection of free curcumin [22] and that CUR-NC and CUR-NCHA nanostructures promotes less tumor growth [23]. For lung cancer (*n* = 2), CUR-P-NPs, CUR-NP (19) CLN-CUR and LN-CUR [19] showed higher tumor growth than free curcumin [14], and reduced tumor weight [14]. SLN-Cur [20] showed the ability to decrease tumor size in Hodgkin’s lymphoma (*n* = 1), while nanostructures such as FA/Nano-Cur allowed a tumor growth inhibition rate of 77.32% compared to free curcumin (16.25%) in the treatment of colon-rectal cancer (*n* = 1) [13]. CUR-MPEG-PLA was shown to be competitive in the treatment of melanoma by reducing tumor volume and weight [15] (*n* = 1).

In all the studies, cancer cell lines were injected in animals by via subcutaneous, but in one the author carried out an intraperitoneal injection and then proceeded with a subcutaneous injection. The route of administration of the treatments was oral gavage (*n* = 3) and intravenous administration (*n* = 8).

### 2.4. Quality and Risk of Bias

The online Robvis tool was used to create the risk of bias analysis image [24]. ARRIVE guidelines were used as a basis from which to analyze Quality and Risk of bias assessment of all the included studies (*n* = 11). Figure 5 summarizes the main classifications of each study based on questions described in Appendix A. Criteria were considered “unclear” when they gave incomplete information and “No information” when the information was not reported.

The study design was well executed in most studies, and some articles mentioned outcome measures (*n* = 11), statistical analysis (*n* = 11), detailed information of cell lines and/or animal models (*n* = 7), clear experimental procedures performed (*n* = 9), and comprehensive results described (*n* = 8). All studies clearly reported on curcumin encapsulation methods (*n* = 11) and investigation of the characteristics of the nanoparticles used (*n* = 11). All studies clearly mentioned treatment time (*n* = 11), route of administration (*n* = 11), curcumin dose (*n* = 11), and presence of control groups (*n* = 11). In addition, all studies investigated anticancer activity based on the effect of Cur-NP and evaluated the phamacokinetics (*n* = 11)

On the other hand, all the studies failed to produce a clear description of researcher blinding (*n* = 11) and one study did not report information on the Ethics statement, which are items required for risk of bias and quality assessment [14], Overall, all the studies (*n* = 11) showed a general low risk of selection bias of reported outcomes.

## 3. Discussion

Curcumin belongs to the category of the most abundant curcuminoids, a polyphenolic compound, derived from the rhizome of *Curcuma longa*, and it has several pharmacological effects, including neuroprotective, antioxidant, anti-inflammatory, and antitumor [17,22,25].

The antitumor effects of curcumin are attributed to molecular mechanisms that alter mainly angiogenesis and cell growth, but also metastasis suppression and cancer cell apoptosis, in addition to altering cancer transcription factors related to cell proliferation, such as NF-B, AP-1, STAT and PPARy [19,26,27].

Moreover, cancer cell proliferation can be altered by downregulating the Ras proteins, PI3K/Akt, upregulating ERK and modulating the Wnt/ß-catenin signaling pathway, arresting the cell cycle of cancer cells in G2/M [19,23,28,29]. In the process of cancer angiogenesis, curcumin inhibits the growth factor VEGF and ensures stability of the extracellular matrix, which downregulates MMP-2 and MMP-9 and upregulates metalloproteinase-1 [19,27,30].

In the articles included in this review, different nanoparticle systems have been investigated to improve CUR efficacy and pharmacokinetics. The nanoparticles were produced by different approaches, using molecules such as poly lactic-co-glycolic acid (PLGA), chitosan, glycerol monooleate (GMO), Gelucire^®^, and polycaprolactone (PCL), among others. A wide variation in nanoparticle size (30.47 to 327.9 nm) and in zeta Potential (−25.0 to 8.1 mV) was noted between the selected studies [7,11,12,13,14,16,18,19,20,22,23]. As reported by Bagheri and co-workers [31], the characteristics of a nanoparticle can interfere in the pharmacokinetic profile of curcumin [31].

Curcumin can be delivered by several routes of administration such as oral, intra venous, intratumoral, nasal, intraperitoneal, topical, or subcutaneous [32]. Herein, the selected studies adopted mainly oral [11,14,20] and intravenous administrations [7,12,13,18,19,22,23] for free and nanostructured curcumin delivery to investigate pharmacokinetic and antitumoral effects.

The encapsulation of Curcumin within nanoparticles provides new, more optimized forms of presentation of this molecule to biological systems. The various possibilities arising from the physicochemical characteristics of nanoparticles can extend the half-life of the molecule and increase its interaction with cancer cells by targeting specific tissues [33,34,35].

Oral administration is considered the most widely accepted route for drug delivery due to good patient compliance, pain avoidance and ease of production. However, curcumin has poor solubility and bioavailability through this route, due to poor intestinal absorption and rapid metabolism [31,32]. The use of nanoparticles can improve these limitations and represents an effective alternative strategy for curcumin delivery [8,13,14]. Selected studies reported that the level of curcumin is higher when incorporated with nanoparticles and presented an increase of bioavailability by oral administration [11,14,20]. The phenomenon was observed with single dose administration [14,20], or with continuous treatment for 14 days [11] at different concentrations (1.25 to 2.5 for pharmacokinetic analysis and 20 mg/kg in the antitumoral evaluation). The nanoparticle ExoCUR (2.5 mg/g) demonstrated the highest increase, of 6-fold, when compared to free curcumin in the brain [11].

The other route of administration mainly used by the selected studies is the intra venous (i.v.) injection, mostly by single injection in the tail vein of animals [12,13,16,19,22,23]. This route allowed rapid distribution of curcumin and rapid effect [32]. Most of the pharmacokinetic analyses were performed on plasma obtained from animals treated intravenously with doses of CUR ranging from 1.25–100 mg/kg [7,12,13,16,18,19,22,23]. Among the studies reviewed, the best performance was achieved with CUR-NSps administered through i.v. at 8 mg/kg. The study reported an increase of T ½ (35.95-fold), AUC0-24 (18.90-fold) and MRT (18.90-fold) when compared to free CUR. Higher CUR concentration was also detected in different organs, such as liver, spleen, kidney, brain, and tumor [7].

In both administration routes, different times were used for pharmacokinetics analysis, ranging from 0.25 h to 720 h. Most studies applied a single injection in the tail vein of animals [12,13,16,18,19,22,23]. Analysis time varied from 0.08 h to 336 h, and most studies evaluated plasma or organs after the first 24 h. Furthermore, the accumulation of curcumin was evaluated in the liver, lung, kidney, and brain [7,11,12,14,18,19] and in the tumor tissues such as cervical [16,22], Hodgkin’s lymphoma [20], Colon-rectal [13] and breast cancer [28].

Different nanoparticle systems have improved CUR efficacy and pharmacokinetics. In general, studies suggest that pharmacokinetic parameters, such as area under the plasma concentration curves (AUC), mean residence time (MRT), blood elimination half-life (T1/2), and Cmax of CUR, are significantly extended following the administration of nanocarried CUR compared to free CUR [7,11,12,13,14,16,18,19,20,22,23].

Tissue distribution of CUR is also enhanced by nanotechnology strategies. For example, a 6-fold higher brain bioaccumulation was reported in healthy Sprague-Dawley rats treated orally with exosomal curcumin than with the free agent [11]. CUR-loaded polymeric NPs preferentially accumulated in the liver without causing degenerative changes, which may be ideal for treating liver disease [12]. Only two studies evaluated the biodistribution of CUR-NPs in tumor-bearing mice [7,19].

Based on passive targeting, all the included studies which used intravenously administered nanoparticle systems resulted in higher CUR content in the tumors compared to the free CUR [7,12,13,16,18,19,22,23]. Indeed, these systems showed distinct tumor selectivity.

Higher concentration of polymer NPs (Cur-P-NPs) was present in the A549 lung tumor, while lower uptake was detected mainly in the lung and liver [19]. On the other hand, curcumin nanosuspensions (CUR-NSps) accumulated in a greater proportion in the spleen and brain than in the H22 liver tumor [7].

The encapsulation of curcumin in nanoparticles allowed a longer plasma circulation time of the molecule [7,13,16,18,20,22,23]. Interestingly, the elimination half-life of nanosuspensions containing curcumin was 35.95-fold longer compared to free curcumin [7]. In the same line, the polymeric nanoparticles showed values for the area under the curve of approximately 498.5 mg/L h, while free curcumin maintained a value of approximately 64 mg/L h [16]. The longer circulation time of curcumin allows for greater uptake at tumor sites, corroborating the results of antitumor activity, which demonstrated a reduction in tumor progression observed with treatments with Cur-NPs [31].

Only one study did not evaluate the antitumor effect [22]. The other studies demonstrated significant inhibition of tumor growth when the animal received treatments with curcumin loaded with nanoparticles dual [7,11,12,13,14,16,18,19,20,23]. Many types of cancer were analyzed, including cervical (11,18), liver (12,14) lung (14,19), colon-rectal [13], breast [23], melanoma [16], and lymphoma [20]. Concerning the frequency of treatment, when the antitumor activity was evaluated, a single injection was applied, daily or on alternate days for 7 to 30 days. In these studies, the dose of nanostructured Cur ranged from 0.8 to 100 mg/kg and was administered by intravenous or via oral [7,11,12,13,14,16,18,19,20,23].

Sun and coworkers [18] demonstrated that curcumin loaded PLGA nanoparticles inhibited up to 75 per cent of tumor growth compared with the curcumin group [11,18]. Similar results in relation to tumor inhibition were demonstrated by Wang and coworkers [16], since they observed a reduction of 79 per cent of melanoma tumor volume after treatment with curcumin loaded with micelles [11,18]. Xie and coworkers [14] and Ji et al. [23] reported that nanocarriers, containing curcumin, reduced about 50 per cent of tumor in relation to free curcumin [14,23]. Another study demonstrates that metalloproteinase-responsive curcumin-loaded nanoparticles inhibited up to 77% of tumor growth [13].

It has also been demonstrated that formulation of curcumin in solid lipid nanoparticles inhibited 50 per cent of Hodgkin’s lymphoma. Around 3× more significant antitumor effect was observed after treatment with nanoparticle-loaded curcumin in cervical [11], and liver cancer [12]. Hong et al. 2017 and Hu et al. [7] observed around 70 per cent of liver tumor inhibition rate (TIR) when the mice were treated with nanoparticle-loaded curcumin, varying from twofold to fourfold higher than free curcumin in both liver and colon-rectal cancer, respectively demonstrated that a curcumin loaded lipidic nanocarrier reduced around 58 per cent of tumor volume of small-cell lung cancer [7,13,14]. Thus, the nanoparticle system resulted in a higher CUR level in the tumors compared to free CUR, and it was responsible for a higher antitumor effect, given that all studies reported results that demonstrated enhanced antitumoral efficacy when curcumin-loaded nanoparticles were used.

The nanometric size of the nanoparticles promotes a greater surface area in contact with the cells, in an inversely proportional manner, and this phenomenon directly impacts the antitumor efficacy of curcumin. Among the nanoparticles evaluated in the studies included in the review, the hydrodynamic size ranged from 30 to 550 nm, and in all studies in which the evaluation of antitumor activity was carried out, the nanoparticles showed greater antitumor efficacy compared to non-nanostructured curcumin, increasing the tumor inhibition rate by up to approximately 70% [7,11,12,13,14,16,18,19,20,23].

Additionally, the coating of nanoparticles with molecules that have an affinity for molecular sites present in cancer cells is capable of directing these nanoparticles to the target tumor, thus increasing the antitumor activity, as observed in the study by Hu et al., 2020, where the coating of Cur-NPs with folic acid boosted antitumor activity in vivo by approximately 50% compared to uncoated Cur-NPs. A similar result was observed in vitro with HeLa cells, in which curcumin encapsulated in iron oxide nanoparticles coated with folic acid showed a higher intracellular uptake rate compared to uncoated Cur-NPs in HeLa cells [13,34].

## 4. Materials and Methods

### 4.1. Registration Protocol

This review was developed in accordance with the Preferred Reporting Items for Systematic Reviews and Meta-Analyses (PRISMA) guidelines. The study was registered in the International Prospective Register of Systematic Reviews (PROSPERO) under registration number: CRD42022288756.

#### 4.1.1. Inclusion Criteria

This systematic review was based on the inclusion criteria of the PICOS approach (Population, Intervention, Comparison, Outcome and Study Design). The significance of PICO in this study is described below.

P (Problem)–Curcumin has low bioavailability and biodistribution time due to its low absorption.

I (Intervention)–Use nanoparticles containing curcumin. C (Comparison)–Encapsulated and non-encapsulated curcumin pharmacokinetics, bioavailability and antitumoral profiles in vivo and ex vivo.

O (Outcomes)-Primary results considered pharmacokinetics, bioavailability and biodistribution of encapsulated and non-encapsulated curcumin by High Performance Liquid Chromatography (HPLC), and the secondary outcome considered the antitumor effect of encapsulated and non-encapsulated curcumin.

(S) Study Design-We considered studies that evaluated

(i) pharmacokinetics and bioavailability; (ii) biodistribution of curcumin and (iii) its antitumor effect when associated or not with nanostructured curcumin in (iv) tumor-bearing animals based on studies that use the method of (v) HPLC. Only studies in English and within a publication period of five years, that is, from 2017 to 2021, were selected.

#### 4.1.2. Exclusion Criteria

Studies were excluded for the following reasons: (i) reviews, letters, personal opinions, book chapters and congress abstracts; (ii) in vitro studies and clinical trials; (iii) use only of free curcumin or curcumin derivatives; (iv) studies that did not use the HPLC technique in their pharmacokinetic analysis; (v) use of Cur-NPs associated with other antitumor drugs and (vi) full paper copy not available (Appendix A).

### 4.2. Information Sources and Research Strategy

Individual search strategies were designed for each one of the following bibliographic databases: Cochrane, LILACS, Embase, MEDLINE/Pubmed, Clinical Trials, BSV region portal, ScienceDirect, Scopus, and Web of Science (Appendix A). The database search started on 23 September 2021 and ended on 1 October 2021.

The bibliographic manager Mendeley^®^ was used to manage the data. This filter was applied by the authors themselves in the Mendeley Reference Manager 2.85.0 software.

#### 4.2.1. Selection Process

After the bibliographic searches, the duplicate references were removed by Mendeley^®^ software. We selected articles based on eligibility criteria established previously. In this way, the studies found were screened in two phases. In the first moment, 8 reviewers (F. S., M.C., A.O., C.S., L. A., W.P., M.G. and T. P.) in pairs, performed the title and abstract reading and evaluated all articles extracted from the databases. In the second stage, all full texts of the articles included were read. The selection of studies was carried out in pairs, with the reading done in the Mendeley software, separating the publications as “included” or “not included”. In the case of disagreement with any study, the reading pair held a discussion about the actual study. And if, even so, there was no agreement, a third author was responsible for deciding whether to include the study in the review. An online form was designed using Google Forms to register the selection process, based on questions related to eligibility criteria. The authors recorded the reason for the exclusion of unselected articles in this form.

#### 4.2.2. Data Collection Process and Quality Analyses

Data extraction was performed in pairs, by the authors M.C. and V.S. and by the pair of L.A. and A.O. The risk of bias analysis was performed using the tools of the Cochrane Collaboration Tool for Bias Risk Assessment. In this way, it was possible to assess the quality of the studies included, through 4 independent and paired reviewers (M.C., V.S., L.A. and A.O.). When there was some divergence of answers, a third reviewer was consulted.

The quality of the experimental design of studies was evaluated with questions that related the conduct of the study to the credibility between exposure and outcome. In this analysis, 21 questions (Appendix A) were used, based on Arrive Guideline 2.0 for studies in vivo [19], which had to be answered within the four options related to the judgment parameters: “LOW” for low risk of bias, “SOME CONCERNS” for moderate risk of bias, “HIGH” if the risk of bias was severe or “NO INFORMATION” for domains without information. Thus, the reviewers’ evaluation for each domain allowed the general quality of each study to be inferred and the respective results of this evaluation were plotted using Robvis tool [24].

## 5. Conclusions

The scientific evidence found in the systematic search carried out in this study showed that the encapsulation of curcumin in nanoparticles optimized its interaction with biological components, ensuring longer plasma circulation time, greater uptake at tumor sites, and, consequently, greater antitumor efficacy compared to free curcumin in in vivo models. In addition, treatments with the different nanoparticles did not show signs of toxicity related to changes in the main organs, body weight, or adverse reactions in animals, which highlights the safety of these drug delivery platforms. The high heterogeneity of the tumor types evaluated in the included studies demonstrates that the presentation of curcumin in nanoparticles maintained antitumor efficacy indices above those of free curcumin, even in the face of the different characteristics of each tumor type, which corroborates other studies that show nanoparticles as promising drug delivery tools for cancer treatment. Given the results obtained in this systematic review, it is possible to conclude that the encapsulation of curcumin in nanoparticles resulted in improvements in pharmacokinetic parameters in in vivo models, with better delivery of curcumin to tumor sites, enabling the promotion of the antitumor activity of the molecule. 

## Figures and Tables

**Figure 1 pharmaceuticals-16-00943-f001:**
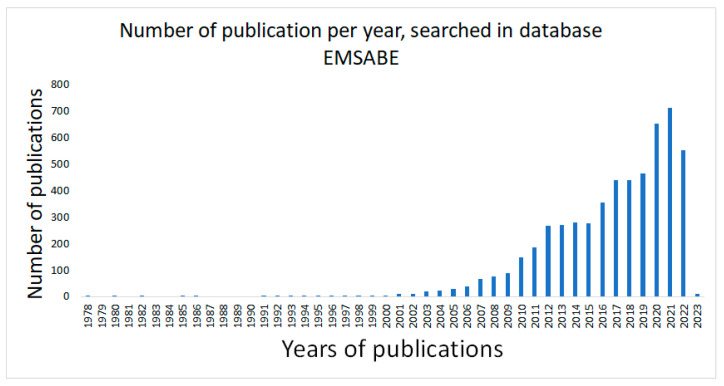
The number of publications per year, searched in the Embase database, refer to search string (TITLE-ABS-KEY ‘curcumin’/exp).

**Figure 2 pharmaceuticals-16-00943-f002:**
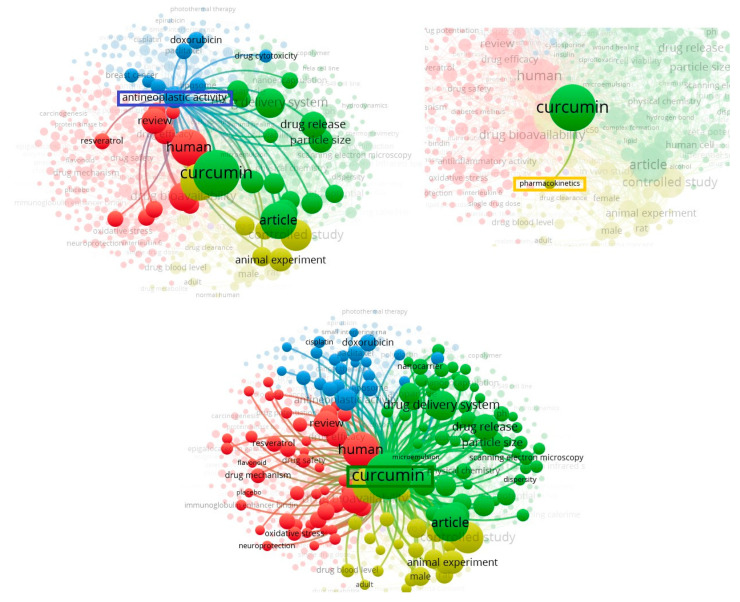
Presentation of the links that the terms “antineoplastic activity”, “pharmacokinetics” and “curcumin” presented in the bibliometric analysis, with the VOSviewer Program version 1.6.17 (21), with the search string (‘curcumin’/exp AND ‘pharmacokinetics’/exp AND ‘antineoplastic agent’/exp) in the title/abstract/keywords, on September 09, 2022, with a minimum co-occurrence of terms of 50 times and “full count”. It was developed using VOSviewer version 1.6.17.

**Figure 3 pharmaceuticals-16-00943-f003:**
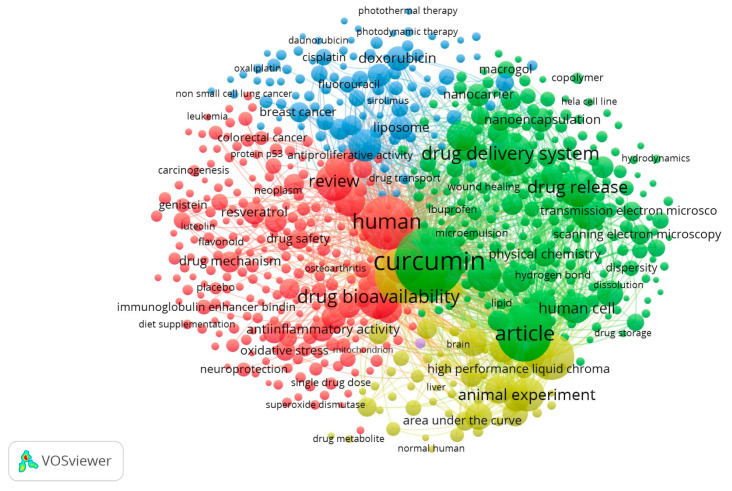
Bibliometric analysis, with the VOSviewer Program version 1.6.17, of the 2752 works published with the search string (‘curcumin’/exp AND ‘pharmacokinet ics’/exp AND ‘antineoplastic agent’/exp) in title/abstract/words-keys, on 9 September 2022, with a minimum co-occurrence of terms of 50 times and “full counting”. It was developed using VOSviewer version 1.6.17(21).

**Figure 4 pharmaceuticals-16-00943-f004:**
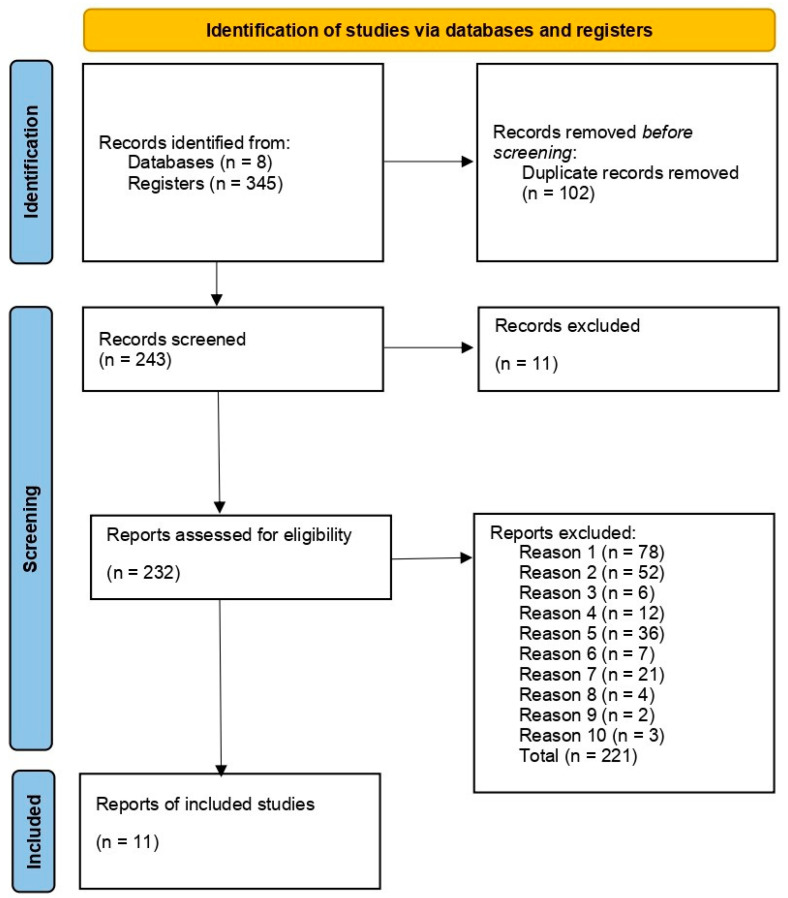
Flow diagram of literature search and selection criteria adapted from PRISMA. The articles that were excluded for certain reasons, numbered from 1 to 10 in “reports excluded” are described in Appendix A.

**Figure 5 pharmaceuticals-16-00943-f005:**
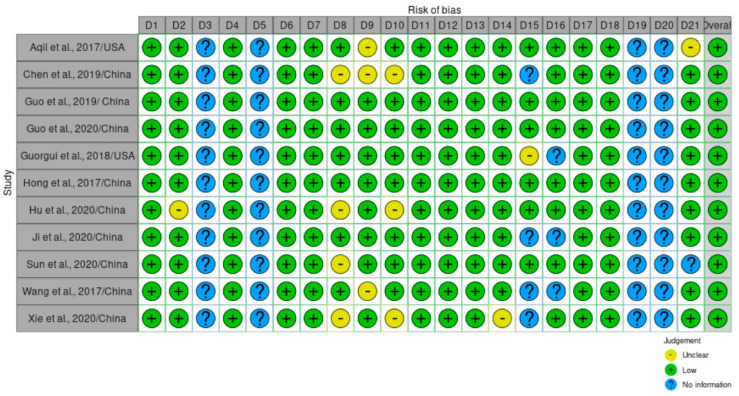
Overall quality of the selected studies. Detailed description of the evaluated parameters is found in Appendix A [7,11,12,13,14,16,18,19,20,22,23].

**Table 1 pharmaceuticals-16-00943-t001:** Summary of descriptive characteristics of the included studies.

Study	Population	Intervention		Outcomes	
Author, Year/Country	Animal Model	Treatment Regimen	Nanostructure Platform	Pharmacokinetic Analysis	Antitumoral and/or Toxicity Analysis
Aqil et al., 2017/USA [11]	Pharmacokinetics analysis: Female Sprague-Dawley rats (*n* = 4) Antitumoral analysis: Athymic nude mice (*n* = 5–8) 5 × 10^6^ CaSki cells/subcutaneous injection	Pharmacokinetics analysis: CUR (2.5 mg/kg) and, ExoCUR (1.25, and 2.5 mg/kg) daily by oral gavage for 14 days. Antitumoral analysis: Exo (80 mg/kg), ExoCUR (CUR-20 mg/kg) and Free CUR (60 mg/kg) by oral gavage 3x/week on alternate days.	Exo: HD: 84 ± 7 nm PDI: 0.19 ± 0.02 ExoCUR: PDI: 0.21± 0.04 HD: 93 ± 6 nm DL: 18–24% EE: 53.9 ± 6.7%	↑ [CUR] in liver (~20 ng/g), lung (~28 ng/g), and brain (~12 ng/g) → (ExoCUR 2.5 mg/kg; *p* < 0.05). ↑ 6-fold [CUR] in the brain → (ExoCUR 2.5 mg/kg)	Cervical tumor xenograft: ↑ 61% of TGI → (ExoCUR; 20 mg/kg; *p* < 0.001) ↑ 21% of TGI → (Exo)
Chen et al., 2019/China [12]	Pharmacokinetics analysis: Kunming strain mice (*n* = 6) Antitumoral analysis: Male mice (*n* = 6) 2 × 10^7^ HCa-F cells/subcutaneous injection on right axilla	Pharmacokinetics analysis: CUR, CPN and CPTN (10 mg/kg) by tail vein single injection. Time of analysis: 0.5, 1, 4, 8, 12, 24 and 36 h. Antitumoral analysis: CUR, CPN and CPTN (10 mg/kg) by tail vein injection daily for 7 days.	CPN: HD: 327.9 ± 14.5 nm ZP: −14.8 ±1.6 mV PDI: <0.21 DL: 6.3 ± 1.1% EE: 71.6 ± 3.4% CPTN: HD: 110.6 ± 2.3 nm ZP: −23.6 ± 2.7 mV PDI: <0.06 DL: 10.1 ± 1.5% EE: 83.2 ± 2.7%	↑ [CUR] in mice livers → (CPTN, 47 µg/g, 4 h) ↑ CUR AUC → (CPTN, 805.75 μg/mL; *p* < 0.05) ↑ CUR liver TI → (CPTN; *p* < 0.05)	Liver cancer: ↓ 31.1% of TV → (CUR) ↓ 56.2% of TV → (CPN; *p* < 0.05) ↓ 84.4% of TV → (CPTN; *p* < 0.05) No impact on mice body weight and no damage to major organs → (CPN and CPTN) ↓ 20% of mice body weight → (CUR)
Guo et al., 2019/China [22]	Pharmacokinetics analysis: Sprague-Dawley rats (*n* = 15) Antitumoral analysis: BALB/c nude female mice 10^7^ MCF-7 cells/subcutaneous injection on left armpit	Pharmacokinetics analysis: Free CUR-DMSO, CUR-NP and CUR-P-NP (1.5 mg/kg) by tail vein single injection. Time of analysis: 8 h, 12 h, and 24 h. Tumor biodistribution analysis: CUR-DMSO, CUR-PCL-NPs and CUR-P-PCL-NPs (0.20 mg/kg of CUR) by i.v. single injection. Time of analysis: 2 h, and 6 h.	CUR-NP: HD: 143.9 ± 0.9 PDI: 0.088 ± 0.024 ZP: −14.6 ± 0.6 DL: 9.65 ± 0.32% EE: 94.37 ± 1.36% CUR-P-NP: HD: 176.9 ± 0.5 PDI: 0.116 ± 0.037 ZP: 8.1 ± 0.7 DL: 7.44 ± 0.16% EE: 87.07 ± 0.63%	1.11% h^−1^ of CUR release at pH = 7.4 → (CUR-P-NP) 0.939% h^−1^ of CUR release at pH = 7.4 → (CUR-NP) ↑ T_1/2_ of CUR → (CUR-P-NP = 0.1229 ± 0.0457 h) ↓ T_1/2_ of CUR → (CUR-DMSO = 0.0743 ± 0.0349 h) ↑ [CUR] on tumor tissue at 2, 6 and 24 h after injection → (CUR-DMSO, CUR-NP and CUR-P-NP, respectively)	
Guo et al., 2020/China [19]	Pharmacokinetics and antitumoral analysis: BALB/c male mice (*n* = 6) 10^7^ A549 cells/subcutaneous injection on each left fore	Pharmacokinetics analysis: Cur-DMSO, Cur-NPs and Cur-P-NPs (50 µg/mL of CUR) by tail vein single injection. Time of analysis: 1 h, 6 h, and 24 h) Antitumor effect: CUR-DMSO, CUR-NPs and CUR-P-NPs (0.8 mg/kg) by tail vein injection every 2 days for 15 days.	CUR-P-NPs: HD: 215 ± 6.183 nm PDI: 0.141 ± 0.013 ZP: 5.90 ± 0.424 mV DL: 7.11% EE: 82.48% Free P-NPs: HD: 184 ± 2.548 nm PDI: 0.139 ± 0.019 ZP: 4.53 ± 0.512 mV	Collagenase IV accelerates CUR release on CUR-P-NPs group. Tumor 1 h → [CUR-DMSO] 0.139 ng/g, [CUR-NPs] 26.524 ng/g and [CUR-P-NPs] 38.490 ng/g (*p* < 0.001) Liver 1 h → [CUR-NPs] (6.267 ng/g) and [CUR-P-NPs] (4.292 ng/g) Lung 1 h → [CUR-NPs] (4.476 ng/g) and [CUR-P-NPs] (15.633 ng/g). Tumor 24 h → [Cur-NPs] (4.448 ng/g) and [Cur-P-NPs] (10.380 ng/g).	Lung cancer: ↑ 19.68% of TGI → (Cur-DMSO); ↑ 66.62% of TGI → (Cur-NPs; *p* < 0.01); ↑ 76.95% of TGI → (Cur-P-NPs; *p* < 0.001); Presence of tumor necrosis → (Cur-NPs and Cur-P-NPs).
Guorgui et al., 2018/US [20]	Pharmacokinetic analysis: BALB/c female mice Antitumoral analysis: SCID female mice (*n* = 8) 5 × 10^6^ L-540 cells/subcutaneous injection	Pharmacokinetics analysis: Free CUR, SLN-CUR and TPGS-CUR (100 mg/kg) by oral single dose. Time of analysis: 0.5, 1, 2 and 4 h. Antitumoral analysis: Free CUR, SLN-CUR and TPGS-CUR (100 mg/kg) orally 5x/week for 18 days.	SLN-CUR: HD: 125.2 nm PDI: 0.268 ± 0.005 ZP: −19.4 ± 2.2 mV TPGS-CUR: HD: 285 nm PDI: 0.283 ± 0.006 ZP: −21.2 ± 2.6 mV	↑ SLN-CUR and TPGS-CUR AUC (1.508 and 1.042 ng/mL, respectively) ↓ Free CUR AUC (231.5 ng/mL) ↑ Tumor accumulation of SLN-CUR and TPGS-CUR (4.5 and 3.0-fold increase, respectively; *p* < 0.05)	Hodgkin’s lymphoma: ↓ 50.5% of TG → (SLN-Cur; *p* < 0.02) ↓ 43% of TG → (TPGS-Cur; *p* < 0.04) No significative difference of TG on TPGS-CUR and SLN-CUR compared to free CUR treatment (*p* = 0.30 and *p* = 0.13, respectively) No toxicity of SLN-CUR and TPGS-CUR.
Hong et al., 2017/China [7]	Pharmacokinetics Sprague male rats (*n* = 10) Antitumoral (*n* = 10) and biodistribution (*n* = 5) analysis ICR mice 10^7^ H22 cells/intraperitoneal injection	Pharmacokinetic analysis: Free CUR and CUR-NSps (10 mg/kg) by i.v. single injection. Time of analysis: 0.08 h, 0.16 h, 0.33 h, 0.5 h, 1 h, 2 h, 4 h, 8 h, 12 h e 24 h. -Biodistribution analysis: Free CUR and CUR-NSps (8 mg/kg) by i.v. single injection. Time of analysis: 0.5, 2, 4, 8, 12, and 24 h. -Antitumoral analysis: Free CUR and CUR-NSps (10 mg/kg) by i.v injection on alternate days for 6 days.	CUR-NSps: HD: 186.33 ± 2.73 nm PDI: 0.22 ZP: −19.00 ± 1.31 mV	↑ T_1/2_ 35.95-fold → (CUR-NSps; *p* < 0.001) ↑ AUC_0–24_ 4.5-fold → (CUR-NSps); *p* < 0.05) ↑ MRT 18.90-fold → (CUR-NSps); *p* < 0.01) ↑ [CUR] on liver, spleen, kidney, brain, and tumor → (CUR-NSps) No detection of plasma [CUR] after 240 min; → (Free-CUR)	Liver cancer ↑ TIR 70.34% → (CUR-NSps; 10 mg/kg; *p* < 0.001) ↑ TIR 40.03% → (CUR; 10 mg/kg; *p* < 0.05) ↑ TIR 55.98% → (CUR- NSps; 5 mg/kg; *p* < 0.05) ↑ TIR 53.21% → (CUR-NSps; 2.5 mg/kg; *p* < 0.05) <5% of hemolysis → (CUR-NSps)
Hu et al., 2020/China [13]	Pharmacokinetic analysis: BALB/C female mice Antitumoral analysis: BALB/C female mice 1 × 10^6^ CT26 cells inoculated at right flank	Pharmacokinetic analysis: Free CUR, Nano-CUR and FA/Nano-CUR (50 mg/kg) by jugular vein single injection. Time of analysis: 5 h, 10 h, 15 h, 20 h, 25 h, and 30 h. Antitumoral analysis: Free CUR, Nano-CUR, and FA/Nano-CUR (50 mg/kg) by i.v. injection every 2 days for 18 days.	FA/Nano-CUR: HD: 30.47 ± 0.65 nm PDI: 0.17 ZP: −3.55 mV DL: 10% EE: 98% DR: 46.32% Nano-CUR: DR: 44.53%	↑ T_1/2_ of 1.58 and 1.42 h → (FA/Nano-CUR and Nano-CUR, respectively) ↓ T_1/2_ of 0.81 h → (Free-CUR) ↑ AUC of 579.1 and 478.6 µg·mL h → (FA/Nano-CUR and Nano-CUR, respectively) ↓ AUC of 62.28 µg·mL h → (Free-CUR)	Colon-rectal cancer: ↑ TIR of 77.32% → (FA/Nano-CUR) ↓ TIR of 20.77 and 16.25% → (Nano-CUR and Free-CUR, respectively) ↓ PCNA index → (Free-CUR, Nano-Cur and FA/Nano-CUR) ↓ Microvessel density → (FA/Nano-CUR) No toxicity in all treatments
Ji et al., 2020/China [23]	Pharmacokinetic analysis: Sprague-Dawley female rats (*n* = 3) Antitumoral analysis: BALB/c mice female (*n* = 5) 10^6^ 4T1 cells/subcutaneous injection on right flank	Pharmacokinetic analysis: Free CUR, CUR-NC, and CUR-NCHA (2 mg/kg) by tail vein single injection. Time of analysis: 0.25 h, 0.5 h, 1 h, 3 h, 6 h, 12 h, and 24 h. Antitumoral analysis: Free CUR, CUR-NC and CUR-NCHA (5 mg/kg) by i.v. injection every 2 days for 10 days.	CUR-NC: HD: 101.4 ± 7.4 nm PDI: 0.33 CUR-NCHA: HD: 161.9 ± 1.7 nm PDI: 0.25 ZP: −25.0 ± 0.8 mV	↑ T_1/2_ of 53.06 ± 18.21 h (CUR-NCHA; *p* < 0.0001) ↓ T_1/2_ of 11.14 ± 1.63 h (Free-CUR) <40% CUR release on pH 7.4 (CUR-NCHA) ~80% CUR release on pH 5.0 (CUR-NCHA)	Breast cancer: Deceleration of TG → (↓ CUR-NC and ↓↓ CUR-NCHA; *p* < 0.05) ↓ TW → (CUR-NCHA) No hemolysis of blood cells → (CUR-NCHA); No damage to major organs → (CUR-NCHA)
Sun et al., 2020/China [18]	Pharmacokinetic analysis: Male Sprague-Dawley rats (*n* = 5) Antitumoral analysis: Balb/c nude xenograft mice 1 × 10^6^ Hela cells/subcutaneous injection on right flank	Pharmacokinetic analysis: Free CUR, CUR-PLGA-NPs and CUR-HPB PLGA-NPs (10 mg/kg) by tail vein single injection. Time of analysis: 0.083, 0.167, 0.33, 0.67, 1, 2, and 3 h. Antitumoral analysis: I.V. administration of CUR (20 mg/kg); PLGA NPs, HPB PLGA NP (10 and 20 mg/kg) every 3 days for 15 days injections by i.v. solution.	PLGA-NPs: HD: 112.8 ± 44 ZP: −6.06 EE: 8.2 ± 0.3% HPB-PLGA NPs: HD: 144.0 ± 44.8 ZP: −1.94 EE: 7.9 ± 0.2%	↑ AUC_0–2_ at 0.08 h → (PLGA NPs) ↑ Plasma maximum [CUR] of 13.29 ± 4.473 µg/mL → (PLGA NPs) ↓ Plasma maximum [CUR] of 1.804 ± 0.256 μg/mL → (Free-CUR) Plasma maximum [CUR] 10.425 ± 2.13 μg/mL → (HPB-PLGA NPs)	Cervical cancer ↑ ~75% TGI → (HPB PLGA NPs; 10 or 20 mg/kg)
Wang et al., 2017/China [16]	Pharmacokinetic analysis: Sprague-Dawley male rats (*n* = 5) Antitumoral analysis: C57BL/6 female mice 10^6^ B16F10 cells/subcutaneous injection on right flank BALB/c female mice 10^7^ A375 cells subcutaneous injection on right flank	Pharmacokinetic analysis: Free CUR and CUR-MPEG-PLA (50 mg/kg) by tail vein single injection. Time of analysis: 120 h, 240 h, 360 h, 480 h, 600 h, and 700 h. Antitumoral analysis: Free CUR and CUR-MPEG-PLA (50 mg/kg)	CUR-MPEG-PLA: HD: 34.5 nm PDI: 0.13 ZP: −2.3 mV EE: 98.8% DL: 10%	↑ 80.6 ± 6.1% CUR release at pH 7.4 after 12 h → (CUR-MPEG-PLA) ↑ AUC: 498.5 mg/L h → (CUR-MPEG-PLA)	B16 melanoma: ↓ 79.4% TV (CUR-MPEG-PLA; *p* < 0.01) ↓ TW (CUR-MPEG-PLA; *p* < 0.01) A375 melanoma: ↓ 68.4% TV (CUR-MPEG-PLA; *p* < 0.01) ↓ TW (CUR-MPEG-PLA; *p* < 0.01) ↓ Number of microvessels (CUR-MPEG-PLA; *p* < 0.05)
Xie et al., 2020/China [14]	Pharmacokinetic analysis: Male rats (*n* = 6) Antitumoral analysis: C57BL/6J male mice (*n* = 6) 10^6^ SCLC H446 cells subcutaneous injection on the left flank	Pharmacokinetic analysis: Free CUR, CLN-CUR and LN-CUR (45 mg/kg) by oral single dose. Time of analysis: 0.5 h, 1 h, 1.5 h, 20 h, 40 h, 60 h, and 80 h. Antitumoral analysis: Free CUR, CLN-CUR and LN-CUR (10 mg/kg) by oral daily dose for 30 consecutively days	CLN-CUR: HD: ~550.1 nm ZP: ~12.7 mV EE: ~79.23%	↑ Gastrointestinal absorption at 1.35 and 1.34-fold compared to free CUR → (CLN-CUR and LN-CUR, respectively; *p* < 0.05) ↑ AUC at 8.94 and 1.38-fold compared to free CUR → (CLN-CUR and LN-CUR, respectively; *p* < 0.05)	Small Cell Lung Cancer: ↓ 58.39% of TV and 59.70% of TW → (CLN-CUR; *p* < 0.01) ↓ 34.43% of TV and 29.21% of TW → (free-CUR; *p* < 0.01) No damage or inflammation in major organs → All treatments

[ ]: Concentration; AUC: Area under the curve CLN-CUR: Polvsaccharide chitosan-cloaked lipidic Curcumin nanoparticles; CPN: Polylactic-co-glycolic acid Curcumin nanoparticles; CPTN: Polylactic-co-glycolic acid and D-a-Tocopheryl polyethylene glycol 1000 succinate Curcumin nanoparticles; CUR: Curcumin; CUR-MPEG-PLA: Curcumin MPEG-PLA micelles; CUR-NC: Curcumin nanocrystals; CUR-NCHA: Hyaluronic acid conjugated Curcumin nanocrystals; CUR-NPs: MePEGTriCL Curcumin nanoparticles; CUR-P-NPs: MePEGPeptideTriCL Curcumin nanoparticles; CUR-PCL-NPs: MePEG-PET-PCL Curcumin Nanoparticle; CUR-PLGA-NPs: Curcumin poly(lactic-co-glycolic acid) nanoparticles; CUR-HPB PLGA-NPs: Hydophobin-coated curcumin poly(lactic-co-glycolic acid) nanoparticles; CUR-P-PCL-NPs: MePEG-peptide-PET-PCL Curcumin nanoparticle; CUR-NSps: Curcumin Nanosuspensions; DL: Drug loading; DMSO: Dimethyl sulfoxide; DR: Drug release; ExoCUR: Exosomal Curcumin; EE: Encapsulation efficiency; Exo: Blank Exosomals; FA/Nano-CUR: Folic acid conjugated MPEG–PCL Curcumin micelle; HD: Hydrodynamic diameter; I.V.: Intravenous injection; LN-CUR: Curcumin lipidic nanoparticles; MPEG–PCL: Methoxypoly (ethylene glycol)-poly(ε-caprolactone); MPEG-PLA: Monomethyl Poly(ethylene glycol) poly (lactide); MRT: mean residence time; Nano-CUR: MPEG–PCL Curcumin micelle; PCNA: Rabbit anti-mouse proliferating cell nuclear antigen; PDI: Polydispersion index; SLN-CUR: Solid Lipid Curcumin nanoparticles; T_1/2_: Half-life time; TG: Tumor growth; TGI: Tumor growth inhibition; TI: Tumor intake; TIR: Tumor inhibition rate; TPGS-CUR: D-a-Tocopheryl polyethylene glycol 1000 succinate Curcumin Nanoparticles; TV: Tumor volume; TW: Tumor weight; ZP: Zeta potential. ↑ increase; ↓ decrease; → in.

## Data Availability

Data available in a publicly accessible repository.

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
