# Peer review of "Pharmacokinetics of Curcumin Delivered by Nanoparticles and the Relationship with Antitumor Efficacy: A Systematic Review"

_pharmaceuticals, 2023, doi:10.3390/ph16070943_

Round 1
Reviewer 1 Report
In this review, the authors found that the curcumin-loaded nanoparticles could improve the stability and bioavailability of the free curcumin (Cur). Furthermore, the after encapsulating by nanopartilces, the anti-tumor efficacies of free Cur was also improve. However, too few references were studied, which means the lack of coverage. Therefore, this review needs a minor revision before accepting.
1. The languages need to be carefully revised. There are many spelling and grammatical mistakes in this manuscript. Worse still, cacography can be found in the title.
2. Only 28 references were studied in this manuscript. It is too few for a systematic review.
3. The relationships between the pharmacokinetics and the antitumor efficacies of the curcumin nanoparticles were not well explained.
Author Response
Reviewer 1
In this review, the authors found that the curcumin-loaded nanoparticles could improve the stability and bioavailability of the free curcumin (Cur). Furthermore, the after encapsulating by nanoparticles, the anti-tumor efficacies of free Cur was also improve. However, too few references were studied, which means the lack of coverage. Therefore, this review needs a minor revision before accepting.
- The languages need to be carefully revised. There are many spelling and grammatical mistakes in this manuscript. Worse still, cacography can be found in the title.
Answer - Extensive proofreading and language was done by a native speaker. The title has been changed to “Pharmacokinetics of curcumin delivered by nanoparticles and the relationship with antitumor efficacy: a systematic review”.
- Only 28 references were studied in this manuscript. It is too few for a systematic review.
Answer –More references have been included in this manuscript.
- The relationships between the pharmacokinetics and the antitumor efficacies of the curcumin nanoparticles were not well explained.
Answer - Thanks for the comment. We have included more information about relationships between the pharmacokinetics and the antitumor efficacies of the curcumin nanoparticles to clarify this aspect.

Reviewer 2 Report
The authors present a systemic review of the Pharmacokinetics of curcumin delivered by nanoparticles and its relationship with antitumor efficacy
The review well written and organized specially the part of material and method. I think it was better if the review included the pharmacokinetic studies achieved using other chromatographic techniques like LC-MS/MS
I suggest changing the title to start with “systemic review”
Author Response
Reviewer 2
The authors present a systemic review of the Pharmacokinetics of curcumin delivered by nanoparticles and its relationship with antitumor efficacy. The review well written and organized specially the part of material and method. I think it was better if the review included the pharmacokinetic studies achieved using other chromatographic techniques like LC-MS/MS. I suggest changing the title to start with “systemic review”.
Answer – Thank you for your comment. Initially, we searched for studies that achieved all chromatographic techniques and then we considered only studies by High Performance Liquid Chromatography (HPLC) for refining. We decided to include only the studies that performed the HPLC technique because the number of studies including the other techniques made the review unfeasible, given that many studies were found.

Reviewer 3 Report
I suggest to change the type of paper from A"article" to "Review".
Generally the authors should add in Introduction some lines on the natural products in health and the importance of network in research approach and related references should be added such as:
Singla, et al (2023). The International Natural Product Sciences Taskforce (INPST) and the power of Twitter networking exemplified through #INPST hashtag analysis. Phytomedicine : international journal of phytotherapy and phytopharmacology, 108, 154520. https://doi.org/10.1016/j.phymed.2022.154520
Some lines on main features of curcumin should be added and related references such as:
Zielińska et al. (2020). Properties, Extraction Methods, and Delivery Systems for Curcumin as a Natural Source of Beneficial Health Effects. Medicina (Kaunas, Lithuania), 56(7), 336. https://doi.org/10.3390/medicina56070336
The novelty character of the research should be better marked.
Results in Figure 5 and Table 5 should be better described in the text.
Author Response
I suggest to change the type of paper from A "article" to "Review".
Answer – Thank you for your observation. In our title and on the submission platform (MDPI) we selected this article as a review. Also, we included the word "review" in the title - “Pharmacokinetics of curcumin delivered by nanoparticles and the relationship with antitumor efficacy: a systematic review”.
Generally the authors should add in Introduction some lines on the natural products in health and the importance of network in research approach and related references should be added such as: Singla, et al (2023). The International Natural Product Sciences Taskforce (INPST) and the power of Twitter networking exemplified through #INPST hashtag analysis. Phytomedicine: international journal of phytotherapy and phytopharmacology, 108, 154520. https://doi.org/10.1016/j.phymed.2022.154520. Some lines on main features of curcumin should be added and related references such as: Zielińska et al. (2020). Properties, Extraction Methods, and Delivery Systems for Curcumin as a Natural Source of Beneficial Health Effects. Medicina (Kaunas, Lithuania), 56(7), 336. https://doi.org/10.3390/medicina56070336/
Answer – Thank you for your contribution and suggestions! These and other references were included in the introduction section and all alterations are highlighted in the text. We described more natural products in health and discussed the main features of curcumin related to properties, extraction methods, and delivery systems.
The novelty character of the research should be better marked.
Answer – We rewrote the text to highlight the novel character of the research.
Results in Figure 5 and Table 5 should be better described in the Text.
Answer – The entire text of the results was rewritten to make the description of the data presented in figure 5 and table 5 clearer.

Reviewer 4 Report
This manuscript reviewed research involving the comparison of the pharmacokinetics (biodistribution, bioavailability and clearance) between free and nanostructured curcumin and their respective relationships with antitumor efficacy. The results have some importance for the following the antitumor-related research of curcumin delivery with nanoparticles. After reading through the manuscript, I have to suggest revision with detailed comments as follows:
1. The manuscript reviewed the difference of antitumor efficacy between free curcumin and curcumin nanoparticles. However, the reasons behind this phenomenon were lacking. The authors should discuss extensively.
2. What is antitumor efficacy of curcumin delivering using different nanoparticles? What is effect of the parameters (such as size, zeta potential and PDI) of these nanoparticles on antitumor ability?
Author Response
Reviewer 4
This manuscript reviewed research involving the comparison of the pharmacokinetics (biodistribution, bioavailability and clearance) between free and nanostructured curcumin and their respective relationships with antitumor efficacy. The results have some importance for the following the antitumor-related research of curcumin delivery with nanoparticles. After reading through the manuscript, I have to suggest revision with detailed comments as follows:
- The manuscript reviewed the difference of antitumor efficacy between free curcumin and curcumin nanoparticles. However, the reasons behind this phenomenon were lacking. The authors should discuss extensively.
Answer - Thank you for your excellent comments and suggestions! We rewrote the text to highlight the difference in antitumor efficacy between free curcumin and curcumin nanoparticles.
- What is antitumor efficacy of curcumin delivering using different nanoparticles? What is effect of the parameters (such as size, zeta potential and PDI) of these nanoparticles on antitumor ability?
Answer - Thank you for your contribution! Certainly, it is very important to describe the effect of physicochemical parameters of free and associated curcumin with different nanoparticles. We rewrote parts of the text to clarify the difference in antitumor efficacy between free curcumin and curcumin nanoparticles.
